# Bridging Brain and Cognition: A Multilayer Network Analysis of Brain Structural Covariance and General Intelligence in a Developmental Sample of Struggling Learners

**DOI:** 10.3390/jintelligence9020032

**Published:** 2021-06-15

**Authors:** Ivan L. Simpson-Kent, Eiko I. Fried, Danyal Akarca, Silvana Mareva, Edward T. Bullmore, Rogier A. Kievit

**Affiliations:** 1MRC Cognition and Brain Sciences Unit, University of Cambridge, Cambridge, Cambridgeshire CB2 7EF, UK; Danyal.Akarca@mrc-cbu.cam.ac.uk (D.A.); Silvana.Mareva@mrc-cbu.cam.ac.uk (S.M.); Rogier.Kievit@radboudumc.nl (R.A.K.); 2Department of Clinical Psychology, Leiden University, 2300 RA Leiden, The Netherlands; eiko.fried@gmail.com; 3Department of Psychiatry, University of Cambridge Clinical School, Cambridge, Cambridgeshire CB2 0SP, UK; etb23@cam.ac.uk; 4Cognitive Neuroscience Department, Radboud University Medical Center, 6525 GA Nijmegen, The Netherlands

**Keywords:** general intelligence, cortical volume, fractional anisotropy, brain structural covariance, cognitive network neuroscience, multilayer network analysis

## Abstract

Network analytic methods that are ubiquitous in other areas, such as systems neuroscience, have recently been used to test network theories in psychology, including intelligence research. The network or mutualism theory of intelligence proposes that the statistical associations among cognitive abilities (e.g., specific abilities such as vocabulary or memory) stem from causal relations among them throughout development. In this study, we used network models (specifically LASSO) of cognitive abilities and brain structural covariance (grey and white matter) to simultaneously model brain–behavior relationships essential for general intelligence in a large (behavioral, N = 805; cortical volume, N = 246; fractional anisotropy, N = 165) developmental (ages 5–18) cohort of struggling learners (CALM). We found that mostly positive, small partial correlations pervade our cognitive, neural, and multilayer networks. Moreover, using community detection (Walktrap algorithm) and calculating node centrality (absolute strength and bridge strength), we found convergent evidence that subsets of both cognitive and neural nodes play an intermediary role ‘between’ brain and behavior. We discuss implications and possible avenues for future studies.

## 1. Introduction

General intelligence, or g ([83]), captures cognitive ability across a variety of domains and predicts a wide range of important life outcomes, such as educational and occupational achievement ([45]), and mortality ([18]). In recent years, methods from network analysis have shed new light on both the cognitive abilities that make up general intelligence ([90]), as well as the brain systems purported to support these abilities ([42]; [79]). For instance, the mutualism model ([89]), inspired by ecosystem models of prey–predator relations, states that the positive manifold ([83]), rather than existing in final form since birth, emerges gradually from the positive interactions among different cognitive abilities (e.g., reasoning and vocabulary) over time (see [57]; [59]). Hence, the positive manifold (and, thus, general intelligence) can arise even from originally weakly correlated cognitive faculties. Therefore, according to the mutualism model (also see [90]), general intelligence can be conceptualized as a complex dynamical system. This paradigm allows us to evaluate general intelligence using the statistical tools of network science ([6]) to estimate the relationships among elements of the system(s) under investigation ([35]; [37]).

For example, new innovations in network psychometrics ([28]) have led to a rapid increase in popularity of behavioral network analysis, especially in psychopathology ([16]; [74]). In this framework, psychological constructs (e.g., mental disorders such as depression) are theorized as complex systems, whereby relationships (edges) between nodes (e.g., item responses on a questionnaire) are estimated using weighted partial correlation networks. The use of partial correlations enables the determination of conditional dependencies among variables, after controlling for the associations among every other node in the network ([28]).

This approach has also recently been used to analyze cross-sectional data on general intelligence. For instance, both [53] ([53]) (N = 1800; age range: 16–89 years) as well as [76] ([76]) (N = 1112; age range: 12–90 years) used a network model approach to analyze data from the WAIS-IV cognitive battery ([93]). Model fit to the pattern of intelligence scores was more consistent with the network model than a latent variable approach (g factor). Furthermore, [64] ([64]), in two separate samples, one the same group of struggling learners as studied here (CALM) but with fewer participants (N = 350), no neuroimaging data, and including tasks not analyzed in this study (e.g., motor speed and tower achievement), observed links between cognitive abilities and learning, especially between mathematics skills and more “domain-general” faculties such as backward digit span and matrix reasoning. Although mutualism is an inherently dynamical theory, therefore requiring longitudinal data to adequately assess, these results are compatible with a cross-sectional interpretation of mutualism’s assumption of (mostly) positive associations among cognitive abilities. Moreover, it must be noted that latent variable ([60]) and network models should not solely be compared using goodness-of-fit indices, but should instead be judged based on “theory compatibility” (see [77]) and the proposed “data-generating mechanism” ([86]).

In addition to psychology, network analysis methods have been widely used in neuroscience to describe the relations among brain regions, ushering in the field of network neuroscience ([10]; [34]). Rather than focusing on individual brain regions in isolation, the brain is conceived (similarly to network psychometrics and mutualism) as a complex system of interconnected networks that facilitate behavioral functions, ranging from sensorimotor control to learning. In this light, several influential studies have revealed pervasive properties of brain networks, such as small-world topology ([8], [9]), modularity ([66]; [84]), and hubs ([85]; [88]), which are nodes (e.g., individual brain regions) that share many connections with other nodes within the brain. Together, these organizational properties of brains enable an economical trade-off between minimizing wiring cost and maximizing efficiency (e.g., information transfer) that enable adaptive behavior ([17]).

Although network approaches have provided unique insights within cognitive neuroscience as well as psychology (e.g., psychopathology and intelligence), few studies have integrated them into a so-called multilayer network paradigm ([15]), which models the relationships among variables simultaneously across time (e.g., days, weeks, months, and years) and/or levels of organization (e.g., behavior and brain variables). Two studies have recently pushed this boundary. [47] ([47]) examined the relations between brain structure (cortical thickness and volume) and depression symptoms. They found (via a partial correlation network model) that certain clusters of brain regions (cingulate, fusiform gyrus, hippocampus, and insula) were conditionally dependent, with a subset of depression symptoms (crying, irritability, and sadness). Secondly, in 172 male autistic participants (ages 10–21 years), Bathelt, Geurts, and Borsboom 2020 used “network-based regression” to estimate the relationship between the unique variance of both the autism symptom network and functional brain connectivity (resting-state fMRI). Moreover, they applied Bayesian network analysis to create a directed acyclic graph between autism symptoms’ sub-scores and their neural correlates. They found that communication and social behavior were predicted by their respective resting-state MRI neural correlates (termed “Comm Brain” and “Social Brain”, respectively).

This study builds on these findings and the recent studies mentioned above, by combining a network psychometrics approach to understand individual differences in cognitive ability (general intelligence) with brain structural covariance networks derived from grey matter cortical volume and white matter fractional anisotropy. In doing so, we created a network of networks, which differs from multiplex (same nodes, different edge types across layers) and multi-slice (same nodes and edge types over time such as in fMRI time-series data) networks (see [15]). The advantages of applying this approach are three-fold and complementary. First, it places the brain and behavior, which often do not map onto each other in a simple and reductionistic one-to-one fashion, into the same analytical paradigm (network analysis using partial correlations). This allows for simultaneous estimations and easier visualizations of potential causal links between cognition and structural brain properties, which to our knowledge, has only been performed in a similar way in two other studies, one involving depression ([47]), the other in autism ([12]). Second, it enables the use of centrality estimates (i.e., strength) and community detection algorithms (i.e., Walktrap) to tease apart major clusters of cognitive abilities and brain regions, which could help to pinpoint potential intervention targets (e.g., using cognitive training and/or transcranial magnetic stimulation). Lastly, it aids in establishing a coherent framework for theory building, which has been lacking in both the neuroscience ([62]) and psychological ([35]) literature. This is accomplished by treating both the brain (algorithmic) and behavior (computational) as equally important levels of analysis to study ([65]), and attempting to more directly translate findings from one level to the other. Ultimately, the hope is that relations between brain–behavior nodes can help identify candidate targets (e.g., nodes that ‘bridge’ the brain and cognition) for future interventions in developmental samples of struggling learners, in particular individuals considered ‘low-performing’ on cognitive ability tasks (e.g., students struggling to learn in school).

## 2. Materials and Methods

### 2.1. Participants

The present cross-sectional sample (behavioral, N = 805; cortical volume, N = 246; fractional anisotropy, N = 165; age range: 5 to 18 years) was obtained from the Centre for Attention, Learning and Memory (CALM) located in Cambridge, UK ([48]). This developmental cohort consists of children and adolescents recruited by referrals for perceived difficulties in attention, memory, language, reading, and/or mathematics problems. A formal diagnosis was neither required nor an exclusion criterion. Exclusion criteria included any known significant and uncorrected problems in vision or hearing, and/or being a non-native English speaker.

Cognitive data were obtained on a one-to-one basis by an examiner in a designated child-friendly testing room. The tasks analyzed in this study comprised a comprehensive array of standardized assessments of cognitive ability, including crystallized intelligence (peabody picture vocabulary test, spelling, single-word reading, and numerical operations), fluid intelligence (matrix reasoning), and working memory (forward and backward digit recall, Mr. X, dot matrix, and following instructions). See Table 1 for task descriptions, relevant citations, and summary statistics (Note: from raw cognitive task scores).

Participants were allotted regular breaks throughout each session. When necessary, testing was split into two separate sessions for participants who did not complete the assessments in a single sitting. A subset of participants also underwent MRI scanning (see below for details). It should be noted that, when compared to age-matched controls, CALM sample participants tend to score lower than their peers. For example, a recent study ([80]) compared the CALM cohort with the NKI-Rockland Sample (see [69]) to assess how cognition and its white matter correlates (fractional anisotropy) differed from childhood to adolescence. In terms of cognitive performance, CALM reliably scored lower than the NKI-Rockland Sample, a ‘typically’ developing cohort, on tasks of crystallized and fluid intelligence (see Level I of Figure 2 from [80]). For more information about CALM and its procedures, see http://calm.mrc-cbu.cam.ac.uk/ (accessed on 8 June 2021).

### 2.2. Structural Neuroimaging: Cortical Volume (CV) and Fractional Anisotropy (FA)

CALM neuroimaging data were obtained at the MRC Cognition and Brain Sciences Unit, Cambridge, UK. Scans were acquired on the Siemens 3 T Tim Trio system (Siemens Healthcare, Erlangen, Germany) via a 32-channel quadrature head coil. T1-weighted volume scans were acquired using a whole brain coverage 3D magnetization-prepared rapid acquisition gradient echo (MPRAGE) sequence, with 1 mm isotropic image resolution. The following parameters were used: Repetition time (TR) = 2250 ms; Echo time (TE) = 3.02 ms; Inversion time (TI) = 900 ms; Flip angle = 9 degrees; Voxel dimensions = 1 mm isotropic; GRAPPA acceleration factor = 2. Diffusion-Weighted Images (DWI) were acquired using a Diffusion Tensor Imaging (DTI) sequence with 64 diffusion gradient directions, with a b-value of 1000 s/mm^2^, plus one image acquired with a b-value of 0. Relevant parameters include: TR = 8500 ms, TE = 90 ms, and voxel dimensions = 2 mm isotropic.

We undertook several procedures to ensure adequate MRI data quality and minimize potential biases due to subject movement. For all participants in CALM, children were trained to lie still inside a realistic mock scanner prior to their scan. All T1-weighted images and FA maps were examined by an expert to remove low-quality scans. Moreover, only data with a maximum between-volume displacement below 3 mm were included in the analyses.

As our grey matter metric, we use region-based cortical volume (CV in mm^3^, N = 246, averaged across contralateral homologues), based on the Desikan–Killiany atlas ([24]) and defined as the distance between the outer edge of cortical grey matter and subcortical white matter ([30]). Tissue classification and anatomical labelling were performed on the basis of the T1-weighted scan using FreeSurfer v5.3.0 software, which is documented and freely available for download online (http://surfer.nmr.mgh.harvard.edu/, accessed on 8 June 2021). The technical details of these procedures are described in prior publications ([22]; [31], [32]). FreeSurfer morphology output statistics were computed for each ROI, and also included cortical thickness and surface area (see Appendix A for analyses involving these two metrics). Based on a recent meta-analysis on functional and structural correlates of intelligence ([11]), as well as a previous longitudinal analysis of the UK Biobank sample (see [58]), we included a subset of 10 cortical volume regions in this study: caudal anterior cingulate (CAC), caudal middle frontal gyrus (CMF), frontal pole (FP), medial orbitofrontal cortex (MOF), rostral anterior cingulate gyrus (RAC), rostral middle frontal gyrus (RMF), superior frontal gyrus (SFG), superior temporal gyrus (STG), supramarginal gyrus (SMG), and transverse temporal gyrus (TTG).

From a subset of our neuroimaging data (see [80]), we also calculated fractional anisotropy (FA, N = 165), a proxy for white matter integrity ([91]). We included 10 regions using the Johns Hopkins University DTI-based white matter tractography atlas (see [49]): anterior thalamic radiations (ATR), corticospinal tract (CST), cingulate gyrus (CING), cingulum (hippocampus) (CINGh), forceps major (FMaj), forceps minor (FMin), inferior fronto-occipital fasciculus (IFOF), inferior longitudinal fasciculus (ILF), superior longitudinal fasciculus (SLF), and uncinate fasciculus (UNC).

All steps to compute regional CV estimation and FA maps were implemented using NiPyPe v0.13.0 (see https://nipype.readthedocs.io/en/latest/, accessed on 8 June 2021). To create a brain mask based on the b0-weighted image (FSL BET; [81]) and correct for movement and eddy current-induced distortions (eddy; [43]), diffusion-weighted images were pre-processed. The diffusion tensor model was then fitted, and fractional anisotropy (FA) maps were calculated using *dtifit*. Images with a between-image displacement > 3 mm were then excluded from subsequent analysis steps. This was completed using FSL v5.0.9. To extract FA values for major white matter tracts, FA images were registered to the FMRIB58 FA template in MNI space using a sequence of rigid, affine, and symmetric diffeomorphic image registration (SyN). This was implemented in ANTS v1.9 ([5]). For all participants, visual inspection indicated good image registration. Binary masks from a probabilistic white matter atlas (thresholded at > 50% probability) in the same space were applied to extract FA values.

We used these region-based measures to study brain structural covariance ([2]), which have been used in cross-sectional and longitudinal designs of cognitive ability in childhood and adolescence (e.g., [82]; see [55] for a recent review of longitudinal studies). Emerging theoretical proposals emphasize the role of networks of brain areas in producing intelligent behavior (e.g., Parieto-Frontal Integration Theory (P-FIT), [52] ([52]) and The Network Neuroscience Theory of Human Intelligence, [7]) rather than individual regions-of-interest (ROIs) in isolation (e.g., primarily the prefrontal cortex). As stated above, we selected 10 grey matter and 10 white matter ROIs based upon combined evidence from a recent meta-analysis ([11]) on associations between functional and structural ROIs and cognitive ability, that further extended the P-FIT theory, but also more recent work performed in two large cohorts, one in longitudinal analysis of the UK Biobank sample (grey matter, [58]) and another in the same (cross-sectional) developmental cohort, although with a smaller behavioral sample size (cognitive data, N = 551; white matter, N = 165, same fractional anisotropy data as the present study; no grey matter data used), as that studied here (see, [80]). See Figure 1 for illustrations of ROIs analyzed in this study.

### 2.3. Network Estimation Methods

All statistical analyses and plots were completed using R ([73]) version 3.6.3 (“Holding the Windsock”). Network estimation was performed using the packages bootnet (version 1.4.3, [27]), igraph (version 1.2.6, [4]), qgraph (version 1.6.5, [29]), and networktools (version 1.2.3, [50]). We used these tools to estimate weighted partial correlation networks, which allowed determination of conditional dependencies among our cognitive and neural variables. For example, in a multilayer network, any partial correlation between node A (e.g., matrix reasoning) and node B (e.g., the caudal anterior cingulate) is one that remains after controlling for the associations among A and B with every other node in the network (e.g., other cognitive abilities and cortical volume ROIs). To estimate these networks, we applied Gaussian Graphical Models (Pearson correlations) using regularization (graphical lasso, see [38]) with a threshold tuning parameter of 0.5 and pairwise deletion to account for missingness. These methods have been widely used to generate sparser networks by penalizing for more complex models, thus decreasing the risk of potentially spurious (e.g., false positive) connections and enabling simpler visualization and interpretation of conditional dependencies between nodes ([26]). We hypothesized that our results would show positive partial correlations (in line with mutualism theory) both within cognitive (e.g., as observed in [64]; [76]) and within neural measures (single-layer networks), as well as between brain–behavior variables in the multilayer networks.

Note that age was included as a node in the estimation procedures (i.e., edge weights, centrality, network stability, and community detection) of all partial correlation networks but was not included in the visualizations of our networks and centrality plots, or in network descriptive statistics (i.e., mean, median, and range of edge weights). For a comparison of the use of age (i.e., included in estimation, regressed out beforehand, or removed from the dataset prior to network estimation), see the Appendix A.

### 2.4. Node Strength Centrality (Single-Layer Networks)

To assess the statistical interconnectedness or connectivity of cognitive and neural nodes relative to their neighbors within our single-layer networks, we estimated node strength, a weighted degree centrality measure calculated by summing the absolute partial correlation coefficients (edge weights) between a node and all other nodes it connects to within the network. Note that our brain structural covariance networks involve ROIs that are not necessarily anatomically connected, preventing certain inferences such as information flow. Nodes were classified as central if the magnitude of their strength z-score was positive and equal to or greater than one standard deviation above the mean. We do not discuss or interpret negative centrality values for our single-layer networks.

### 2.5. Community Detection and Bridge Strength Centrality (Multilayer Networks)

In our multilayer networks, we applied the Walktrap community detection algorithm ([72]) to determine in a data-driven manner whether clustering, or grouping, of nodes (e.g., cognitive and/or neural) occurred. The Walktrap algorithm assesses how strongly related nodes are to each other (that can be due to similarity, e.g., because nodes A and B are similar, or it can be because nodes A and B are different, but node A has a strong impact on node B; see “*Topological overlap and missing nodes*” of [36]). The Walktrap algorithm works by taking recursive random walks between node pairs and classifies communities according to how densely connected these parts are within the network (wherever the random walks become ‘trapped’). Walktrap is widely used in the network psychometrics literature and, in a Monte Carlo simulation study, was shown to outperform other algorithms (e.g., InfoMap) for sparse count networks (e.g., those used in diffusion tensor imaging), although it must be noted that this result was for networks made up of 500 nodes or higher ([39]). We also calculated the maximum modularity index value (Q), which estimates the robustness of the community partition ([68]). We interpreted values of 0.5 or above as evidence for reliable grouping.

Instead of traditional absolute strength, we calculated bridge strength, a novel weighted degree centrality measure originality developed to study comorbidity between mental disorders (see [51] for overview). Bridge strength centrality sums the absolute value of every edge that connects one node (e.g., matrix reasoning) in one pre-assigned community (e.g., cognition) to another node (e.g., caudal anterior cingulate) in another pre-assigned community (e.g., brain). Recent simulation work has shown that the method can reliably recover true structures of bridge nodes in both directed and undirected networks ([51]). Rather than relying on straightforward ‘brain’ or ‘behavior’ assignments to classify nodes, we pre-assigned communities for bridge strength calculation based on results from the Walktrap algorithm.

The presence of bridges between communities (e.g., if nodes from topological distinct clusters such as cognition vs. brain feature relations) might suggest the existence of intermediate endophenotypes ([33]; [56]), and potentially identify potential nodes (both cognitive and neural) that might one day guide intervention studies. Nodes were classified as central if the magnitude of their strength z-score was positive and equal to or greater than one standard deviation above the mean. We do not discuss or interpret negative centrality z-score values for our multilayer networks.

### 2.6. Node Centrality Stability (Single and Multilayer Networks)

Lastly, we quantified the reliability of our centrality estimates for all single-layer (absolute strength of cognitive and brain structural covariance nodes) and multilayer networks (bridge strength). We estimated the correlation stability (CS) coefficient, calculated as the maximum proportion (out of 2000 bootstraps) of the sample that can be dropped out and, with 95% probability, still retain a correlation of 0.7 (correlation between rank order of centrality in network estimated on full sample with order of subsampled network in smaller N), with a CS value of 0.5 considered to be stable ([28]). Lastly, also using bootstrapping, we determined the stability of the edge-weight coefficients, but present these results in the Appendix A.

## 3. Results

### 3.1. Single-Layer Network Models (Cognitive, Cortical Volume, and Fractional Anisotropy)

The regularized partial correlation (PC) network for the CALM cognitive data is shown in Figure 2 (top left). This network shows that all partial correlations are positive, and most have small magnitude (mean PC = 0.08, median PC = 0.07, PC range = 0–0.63). One edge (between reading and spelling) was an outlier (PC = 0.63, all others are between 0 and 0.27), likely due to close content overlap (verbal ability). Regarding centrality, three nodes emerged as strong (positive z-score at or greater than one standard deviation above the mean): (in descending order of centrality strength) reading, numerical operations, and peabody picture vocabulary test (Figure 2, top right). Overall, centrality estimates were stable, indicated by a high correlation stability (CS) coefficient of 0.75, revealing that at least 75% of the sample could be dropped while maintaining a correlation of 0.7 with the original sample at 95% probability.

Next, we estimated the partial correlation network among 10 grey matter regions, as shown in Figure 1 (top) above. All edges weights (mean PC = 0.09, median PC = 0, PC range = −0.15–0.52) of the cortical volume network (Figure 2, middle left) were positive, apart from one negative path (caudal middle frontal gyrus and frontal pole PC = −0.15). Note that the negative path between the caudal middle frontal gyrus and frontal pole might be due to the frontal pole correlating surprisingly weakly with other grey matter nodes and displaying a steeper decrease pattern across age (see Figure 3 below and Appendix A). Two ROIs emerged as central (in descending order of centrality strength): superior temporal gyrus and rostral middle frontal gyrus (Figure 2, middle right). Similar to the cognitive network, cortical volume centrality was stable (CS coefficient = 0.52), indicating that about 52% of the sample could be subtracted to maintain a correlation of centrality estimates above 0.7 compared to the full sample. This finding is despite the lower sample size compared to the behavioral data (N = 805 for behavior vs. N = 246 for cortical volume).

Finally, similar to the cognitive and the grey matter covariance network, the fractional anisotropy network (Figure 2, bottom left) has positive partial correlations, with all edge weights varying between small and moderate values: mean PC = 0.08, median PC = 0, and PC range = 0–0.44. Two white matter ROIs displayed centrality (Figure 2, bottom right). These included (in descending order) the forceps minor and inferior longitudinal fasciculus. Finally, fractional anisotropy centrality was moderately stable (CS coefficient = 0.44), indicating that about 44% of the sample could be removed while maintaining a 0.7 association with 95% probability. This is possibly due to the much lower sample size (N = 165) compared to the cognitive (N = 805) and grey matter (N = 246) networks.

### 3.2. Bridging the Gap: Multilayer Networks

For a correlation plot of cognitive tasks and neuroimaging measures, see Figure 3. The regularized partial correlation network analyses for the CALM multilayer networks data are shown in Figure 4. Consistent with the pattern found in the single-layer networks, the cognitive and grey matter multilayer network (top left of Figure 4) edges are mostly positive and small to moderate weights (mean PC = 0.04, median PC = 0, PC range = −0.12–0.64). Comparably, the cognitive and white matter multilayer network (Figure 4, top right) had similar edge weight estimates (mean PC = 0.04, median = 0, range = −0.2–0.65). Finally, combining all measures together (tri-layer network consisting of cognition, grey and white matter, bottom center of Figure 4) produced a network with similar characteristics to the bi-layer networks (mean PC = 0.02, median PC = 0, PC range = −0.2–0.66). For the bi-layer networks, the Walktrap algorithm produced either three (cognition-white matter) or four (cognition-grey matter) clusters that consisted entirely of cognitive or neural nodes, except for following instructions (Ins), which was either kicked out (cognition-grey matter network, Q = 0.56, indicating strong modularity) or grouped with a neural node (forceps minor of the cognition-white matter network, Q = 0.39, indicating moderate modularity). The result for the tri-layer network (Q = 0.25, indicating weak modularity) was more complex, with a total of 15 communities (Figure 4, bottom center; note, age was found to be in a community by itself but is not shown in the figure).

Regarding centrality, we report bridge strength (Figure 5). In the cognitive-grey matter network, three bridge nodes surfaced (in descending order: superior temporal gyrus, superior frontal gyrus, and rostral middle frontal gyrus, Figure 5, top left). In terms of stability, the CS coefficient was 0.20, indicating that the bridge strength estimates were unstable under bootstrapping conditions. In the cognitive-white matter bi-layer network, three nodes (in descending order: uncinate fasciculus, inferior frontal-occipital fasciculus, and hippocampal cingulum) emerged as possible bridge nodes (Figure 5, top right). Moreover, the centrality estimates had a CS coefficient of 0.13, once again suggesting that the bridge strength estimates were unstable. Lastly, for the tri-layer network, five nodes displayed positive bridge strength equal to or greater than one standard deviation above the mean (Figure 5, bottom center). These included (in descending order): reading, peabody picture vocabulary test, superior frontal gyrus, spelling, and numerical operations. Much better than the bi-layer networks, the tri-layer network bridge strength estimates were moderately stable (CS coefficient = 0.44).

## 4. Discussion

### 4.1. Summary of Main Findings

In this study, we used network analysis (partial correlations) to examine the neurocognitive structure of general intelligence in a childhood and adolescent cohort of struggling learners (CALM). For our single-layer networks (Figure 2), we found that cognitive, grey matter, and white matter networks contained mostly (if not all) positive partial correlations. Moreover, in all single-layer networks, at least two nodes emerged as more central than others (as indexed by node strength equal to or greater than one standard deviation above the mean), which varied in stability from moderately to highly reliable. In the cognitive network, this included verbal ability (specifically reading and peabody picture vocabulary test) and crystallized intelligence (i.e., numerical operations). In the structural brain networks (grey matter cortical volume and white matter fractional anisotropy), two nodes passed the centrality threshold, for both the grey matter network (superior temporal gyrus and rostral middle frontal gyrus) and white matter network (forceps minor and inferior longitudinal fasciculus). Furthermore, we extended previous approaches by integrating networks of structural brain data with a cognitive network, forming bi- and tri-layer networks (Figure 4). In doing so, we observed multiple (both positive and negative) partial correlations between brain and behavior variables. Using bridge strength as a metric, we found that, in our bi-layer networks, only neural nodes harbored significant connections across communities (defined by the Walktrap algorithm) and levels of organization (Figure 5, top). In contrast, in the tri-layer network, we found support that mostly cognitive nodes connected across different communities (Figure 5, bottom). Overall, our results suggest which behavioral and neural variables have greater (possible) influence among or might be more influenced by other nodes and potentially serve as bridges between the brain and cognition within general intelligence. However, the literature on drawing inferences from networks to the most likely consequences of intervening on the network is complex and rapidly changing, (e.g., [21]; [46]; [63]).

### 4.2. Interpretation of Network Models and Community Detection Analyses

For the cognitive network, each node corresponded to a single cognitive task (e.g., matrix reasoning), while partial correlations (weighted edges) between nodes were interpreted as compatible with (possible) causal consequences of interactions among cognitive abilities during development. The existence of only positive edges in our cognitive network would be expected under a mutualistic perspective (i.e., interactions among cognitive variables), although longitudinal analyses are needed to further substantiate this claim. Mutualism, which at its essence is a network theory of general intelligence ([89]), hypothesizes that the positive manifold and general intelligence ([83]) emerge from causal interactions among abilities rather than a general latent factor ([35]; [53]). Hence, cognition is viewed as a complex system derived from the dynamic relations of specific abilities that become more intertwined over development. Initially, it was surprising that two of the three most central nodes (i.e., reading and peabody picture vocabulary test) relate to verbal ability rather than abilities such as fluid intelligence and working memory (matrix reasoning and (forward and backward) digit recall), which are traditionally viewed as causal influences on cognitive development ([20]). However, an emerging body of literature suggests that verbal ability plays a crucial role in cognitive development (e.g., between reading and working memory before 4th grade, [70] and [95]), as well as driving the emergence of reasoning ([59]; also see [40]).

As for our neural networks (here, grey matter cortical volume and white matter fractional anisotropy), individual nodes were comprised of a single ROI. Importantly, we did not interpret weighted edges as an index of direct connectivity. Instead, the presence of strong associations between these ROIs would be compatible with the hypothesis of coordinated development (see [2]), whereby certain brain regions show preferential correlations to each other than more peripheral regions over time (e.g., childhood to late adolescence), as well as the notion of “rich” ([87]) and “diverse” ([13]) clubs that enable local and global integration. The most central grey matter node, the superior temporal gyrus, has been implicated in verbal reasoning (e.g., [54]). Regarding white matter, the two strongest nodes (forceps minor and inferior longitudinal fasciculus), while not anatomically close, instead represent long-range connections (see [23]) and have been linked to mathematical ability ([67]) and visuospatial working memory ([61]).

Finally, we integrated both domains (cognitive abilities and brain metrics) into combined multilayer networks (cognition-grey matter, cognition-white matter, and cognition-grey and white matter). Doing so allowed us to attempt comparison and integration simultaneously across explanatory levels within the same analytical paradigm (network analysis) and statistical metrics (partial correlations, centrality, and community detection). From this analysis, three observations immediately stood out. First, there were multiple partial correlations between cognitive and neural nodes (especially in the cognitive-white matter and cognitive-grey matter and white matter networks). Second, compared to the single-layer networks, the multilayer networks have more negative partial correlations. Together, these two findings further suggest that associations between the brain and cognition are complex as they defy straightforward (e.g., only positive and/or one-to-one) relationships and interpretations. However, it should be noted that causality (e.g., conditioning on colliders, see [75] ([75]) for overview of interpretations of correlations in graphical causal models in observational data) becomes even more difficult to determine with networks incorporating multiple levels of organization (e.g., cognition and structural brain covariance). Finally, we found a peculiar role of the cognitive task following instructions (Ins) within all multilayer networks. For example, in the cognitive-grey matter network, Ins had no partial correlations with any other nodes within the network, while in both the cognitive-white matter and tri-layer network (cognition, grey and white matter) Ins only correlated with the forceps minor (FMin), a neural node, and not any of the cognitive variables. This might suggest that following instructions, traditionally a working memory task and often analyzed using structural equation modeling, may have distinct psychometric properties (e.g., one-to-one mapping) when compared to other cognitive tasks when modeled through network science approaches, and/or when adjusted for all shared correlations.

Further inspection of bridge strength centrality showed an interesting pattern: (discounting the one standard deviation cutoff) the neural nodes are stronger than the cognitive variables within the multilayer networks, despite there being an equal number of cognitive nodes for each brain metric. This is possibly due to the large number of edges between them (grey and white matter regions) and both cognitive and other neural nodes. In other words, since the neural nodes contain a larger number of connections (partial correlations) across explanatory levels, they display greater bridge strength (bridge strength sums inter-network correlations).

In other ways, the multilayer networks differed. First, in the tri-layer network, four of the five central nodes were cognitive variables, while, in the bi-layer networks, the central nodes were neural ROIs. Three of these central cognitive nodes in the tri-layer network (reading, peabody picture vocabulary test, and numerical operations) were also found to be central in the single-layer cognitive network. This further suggests the importance of mathematical and verbal ability in understanding the cognitive neuroscience of general intelligence. Secondly, the fact that cognitive nodes were found to be central only in the tri-layer network suggests that grey and white matter, while related, possibly reveal unique information about cognition when combined and analyzed together simultaneously.

### 4.3. Limitations of the Current Study

This study contains several limitations that require caution when interpreting the results. First and foremost, these findings are based on cross-sectional data. While adequate to help tease apart individual differences in cognition between people, cross-sectional data cannot be used to elucidate differences in changes within individuals over time, such as during development. Therefore, longitudinal analyses are needed before attempting to make strong inferences about the dynamics of these networks. Reiterating this point, a recent study using intelligence data ([78]) found that a cross-sectional analysis of the g factor of cognitive ability was unable to capture within-person changes in cognitive abilities over time. This finding further stresses the necessity to integrate cross-sectional (between-person) differences and longitudinal (within-person) changes when studying cognition.

Moreover, the CALM sample represents an atypical sample ([48]), with participants who consistently score lower on measures of attention, learning, and/or memory than age-matched controls (see Figure 2 (Level I) of [80] for comparison to a typically developing sample). As a result, these analyses would need to be replicated in additional (ideally larger) samples with different cognitive profiles before our results can be generalized. This shortcoming of the present study is echoed by the low stability estimates found for the centrality values in the bi-layer networks, which might be due to the differences between the sample sizes of the neural data (grey matter, N = 246; white matter, N = 165) compared to cognition (N = 805). While these discrepancies could affect the statistical power of our results, the amount of neural data used in the present study is considerably larger than the sample sizes commonly used in standard neuroimaging studies ([71]). However, given that the tri-layer network showed moderate bridge strength stability but also displayed weak modularity, and the Walktrap algorithm produced 15 communities in the network, which contained only 31 nodes (including age), we strongly suggest that our results should be interpreted with caution and advise that future studies should aim to analyze neuroimaging data from larger cohorts (e.g., ABCD study, [19]).

Lastly, we re-ran our analyses to test the sensitivity of our main findings (e.g., positive partial correlations and central nodes) to potential outliers (defined as ±4 standard deviations). Doing so did not severely alter the partial correlation weights between nodes in our networks (see Appendix A for detailed comparisons). It must be restated that our data come from an atypical sample, which might influence brain metrics even with rigorous quality control procedures. Therefore, despite this discrepancy, our data supports brain–behavior ‘bridges’ in general intelligence.

### 4.4. Future Directions toward Theory Building in Cognitive Neuroscience

Our results that suggest verbal abilities rather than fluid intelligence or working memory might play a more pivotal role in the development of cognitive ability fits with the gradual progression in schooling. For example, before children can successfully be taught more advanced subjects (e.g., history, reading comprehension, etc.), they must first become competent in basic language faculties. In other words, it may be that verbal skills (e.g., reading and spelling) facilitate performance on abstract tests, even in the absence of direct knowledge-based task demands. Recent evidence has been found supporting this notion and suggest that verbal ability, particularly reading and vocabulary in relation to working memory and reasoning, might drive early cognitive development ([59]; [70]; [95]). Therefore, future studies could further examine whether greater verbal ability in early development facilitates greater acquisition of higher-level cognitive skills by lowering computational demands in working memory.

Moreover, in this context, the fact that the numerical operations task was also found to be central (tri-layer network only) should be expected since mathematics (e.g., arithmetic) also involves symbol manipulation. In terms of mutualism ([89]), future models (ideally in longitudinal samples) could test whether language and other symbolic abilities show progressively higher reciprocal associations during early development compared to other abilities until more complex cognition (i.e., fluid reasoning and working memory) develops in later childhood (also see [59]; [70]).

We argue that future studies should aim to incorporate data from different scales, not only temporal (e.g., development) but also levels of organization (e.g., brain and behavior). Furthermore, results from different levels can more easily be interpreted if these datasets are analyzed and interpreted using a unified quantitative and conceptual framework, such as network science. Last, and perhaps most important, cognitive neuroscientists must formulate mechanistic (e.g., [14]) and generative models (for instance, [1]) to gain further insights from the past and help guide future controlled experiments.

One proposal attempting to explain general intelligence using network neuroscience is The Network Neuroscience Theory of Human Intelligence (NNTHI, [7]). Barbey argues that general intelligence arises from the dynamic small-world typology of the brain, which permits transitions between “regular” or “easy-to-reach” network states (needed to access prior knowledge for specific abilities) and “random” or “difficult-to-reach” (required to integrate information for broad abilities) network states (i.e., as in network control theory, see [44]). Together, this constrained flexibility allows the brain to adapt to novel cognitive domains (e.g., in abstract reasoning) while still preserving access to previously learned skills (e.g., from schooling).

Evidence supporting the NNTHI has been inconclusive so far ([42]). However, two recent studies, although not directly testing the NNTHI, have shed light on the network neuroscience of cognition. [14] ([14]) found that a mechanistic model assuming that “connector hubs” (diverse club nodes, see [13]), which regulate the activity of their neighboring communities to be more modular but maintain the capability of “task-appropriate information integration across communities”, significantly predicted higher cognitive performance on various tasks, including language and working memory. Furthermore, in the same sample studied here, [1] ([1]) applied a generative network modeling approach to simulate the growth of brain network connectomes, finding that it is possible to simulate structural networks with statistical properties mirroring the spatial embedding of those observed. The parameters of these generative models were shown to correlate with neuroimaging measures not used to train the models (including grey matter measures), cognitive performance (including vocabulary and mathematics), and relate to gene expression in the cortex.

Together, these studies point the field toward a better mechanistic understanding of the development of human brain structure, function, and their relationship with cognitive ability. Researchers must not shy away from but rather embrace the complexity of the brain and cognition (see [37] for a similar argument for mental health research). Intelligence is a complex system—to understand it, we must treat it as such.

## Figures and Tables

**Figure 1 jintelligence-09-00032-f001:**
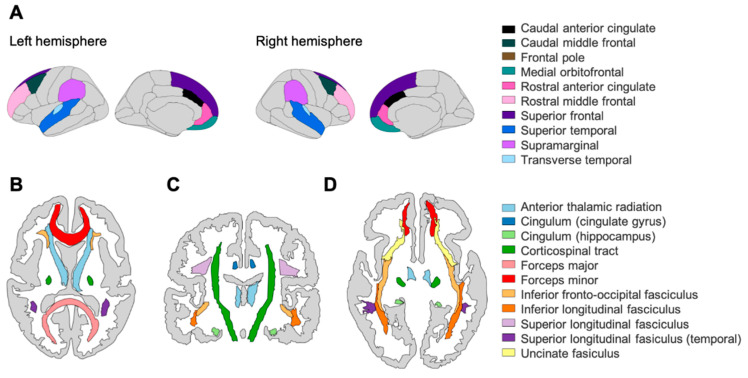
(**A**) Grey matter ROIs based on the Desikan–Killiany atlas (cortical volume, N = 246) in the left and right hemisphere. White matter ROIs based on the John’s Hopkin’s University atlas (fractional anisotropy, N = 165) in (**B**) transverse plane (superior), (**C**) coronal plane, and (**D**) transverse plane (inferior). Note that the frontal pole is not visible in these planes.

**Figure 2 jintelligence-09-00032-f002:**
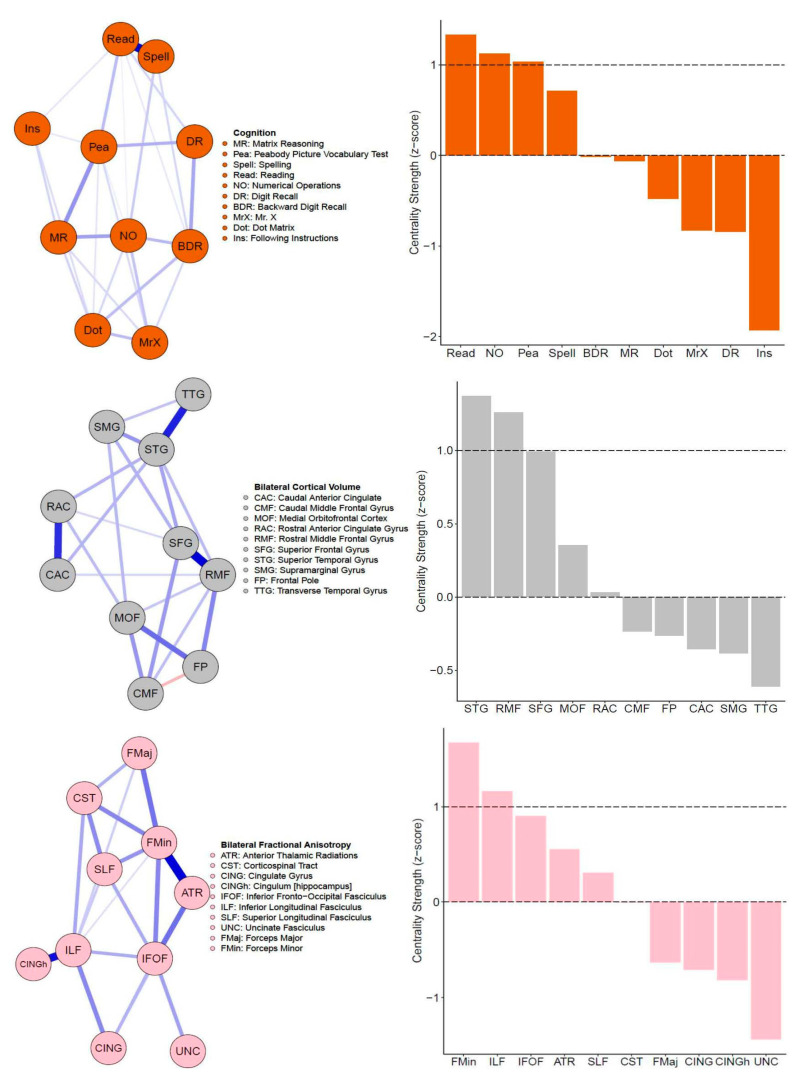
Single-layer partial correlation networks. Top: Network visualization (spring layout, **left**) of CALM cognitive data (N = 805). Centrality estimates (z-scores) of all cognitive tasks (**right**). Middle: Network visualization (spring layout, **left**) of CALM cortical volume data (N = 246). Centrality estimates (z-scores) of all cortical volume nodes (**right**). Bottom: Network visualization (spring layout, **left**) of CALM fractional anisotropy data (N = 165). Centrality estimates (z-scores) of all fractional anisotropy nodes (**right**). Dashed lines in centrality plots indicate mean strength and one standard deviation above the mean.

**Figure 3 jintelligence-09-00032-f003:**
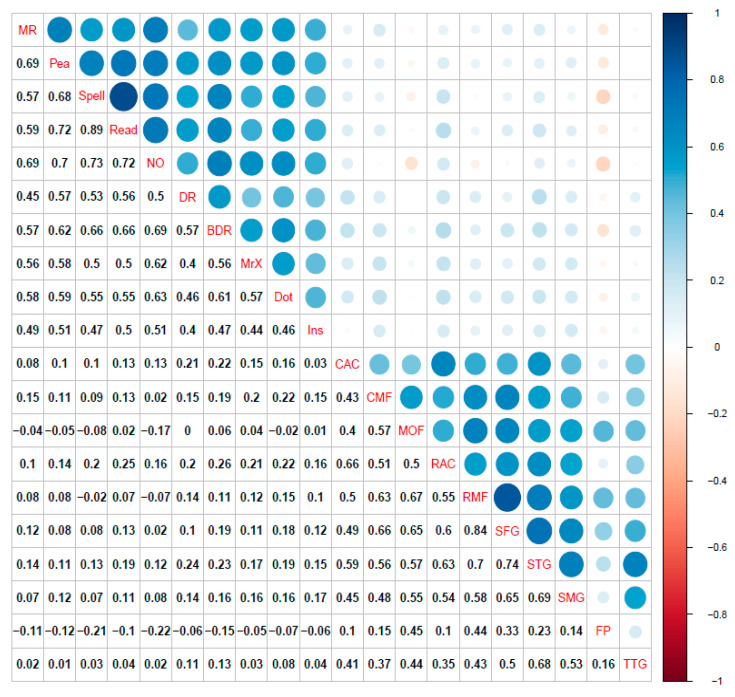
(**Top**) Correlation plot for cognitive raw scores and bilateral cortical volume ROIs. (**Middle**) Correlation plot for cognitive raw scores and bilateral fractional anisotropy ROIs. (**Bottom**) Correlation plot for bilateral cortical volume and bilateral fractional anisotropy ROIs. All coefficients shown are Pearson correlations. Blue represents positive correlations while red signifies negative correlations among variables. Size of circles indicates the magnitude of the association (e.g., larger circle = higher correlation). Correlations calculated using pairwise complete observations. Abbreviations: matrix reasoning (MR), peabody picture vocabulary test (Pea), spelling (Spell), single word reading (Read), numerical operations (NO), digit recall (DR), backward digit recall (BDR), Mr. X (MrX), dot matrix (Dot), following instructions (Ins), caudal anterior cingulate (CAC), caudal middle frontal gyrus (CMF), medial orbital frontal cortex (MOF), rostral anterior cingulate gyrus (RAC), rostral middle frontal gyrus (RMF), superior frontal gyrus (SFG), superior temporal gyrus (STG), supramarginal gyrus (SMG), frontal pole (FP), transverse temporal gyrus (TTG), anterior thalamic radiations (ATR), corticospinal tract (CST), cingulate gyrus (CING), cingulum (hippocampus) (CINGh), inferior fronto-occipital fasciculus (IFOF), inferior longitudinal fasciculus (ILF), superior longitudinal fasciculus (SLF), uncinate fasciculus (UNC), forceps major (FMaj), and forceps minor (FMin).

**Figure 4 jintelligence-09-00032-f004:**
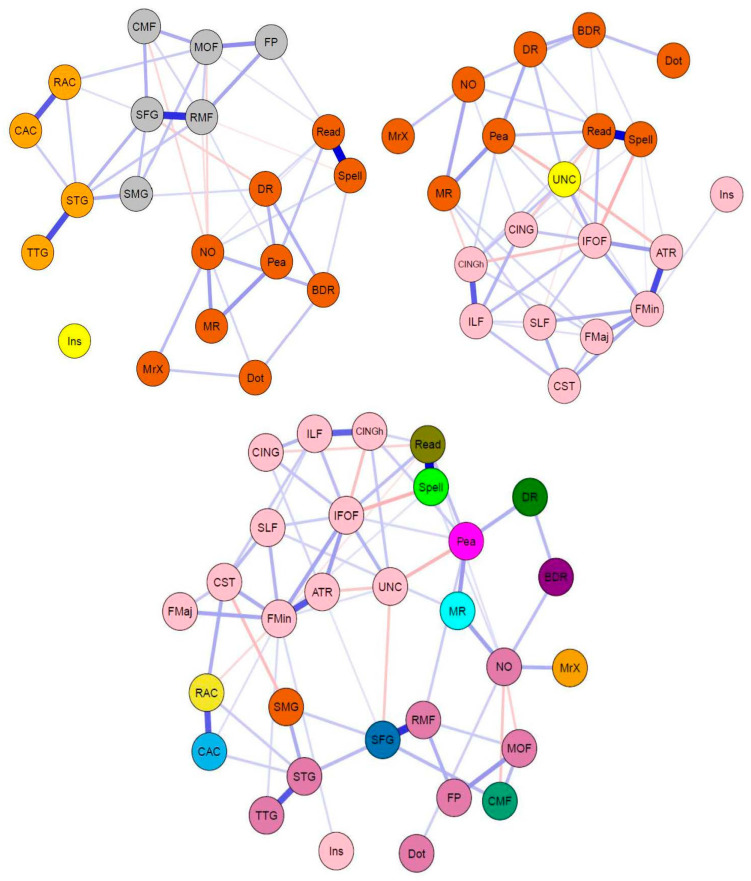
Network visualizations (spring layout) of partial correlation multilayer networks for CALM data. Colors indicate groups determined by the Walktrap algorithm (see above). (**Top**) Bi-layer networks consisting of cognition and grey matter (**top left**), and cognition and white matter (**top right**). (**Bottom**) Tri-layer network consisting of cognition, grey matter, and white matter (**center**).

**Figure 5 jintelligence-09-00032-f005:**
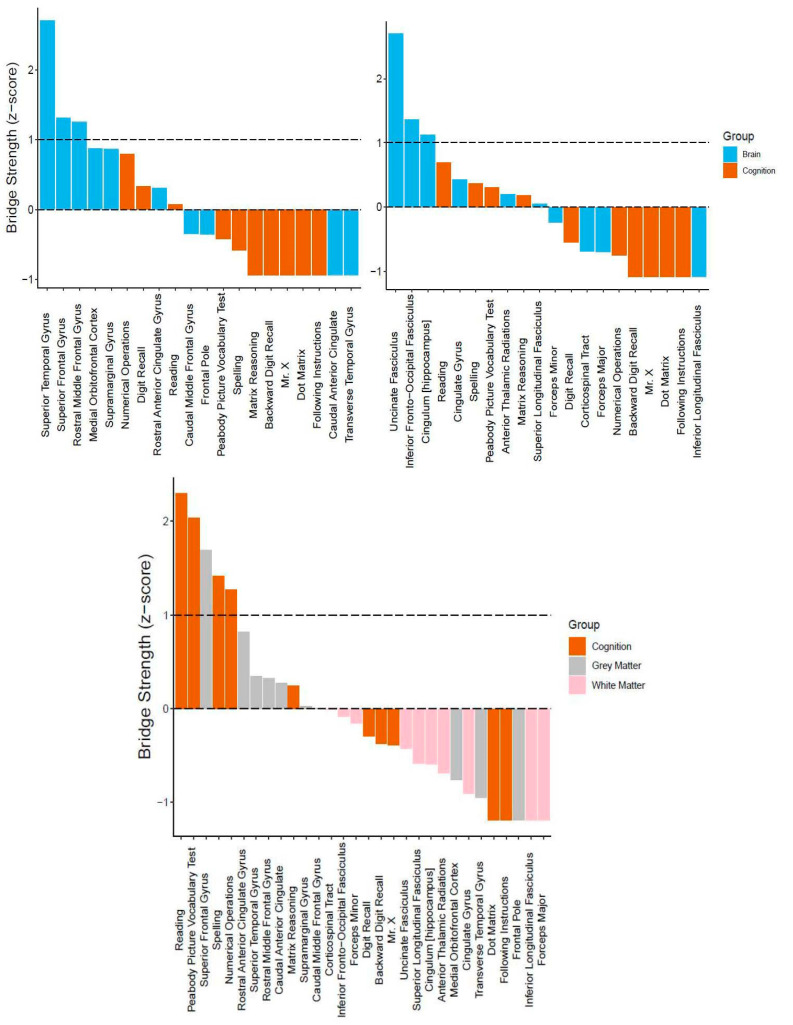
Bridge centrality estimates (z-scores) for multilayer networks. (**Top**) Bi-layer networks consisting of cognition and grey matter (**top left**), and cognition and white matter (**top right**). (**Bottom**) Tri-layer network consisting of cognition, grey matter, and white matter (**center**). Dashed lines indicate mean strength and one standard deviation above the mean.

**Table 1 jintelligence-09-00032-t001:** List, descriptions, and summary statistics (mean, standard deviation, range, and percentage of missing data) of cognitive assessments used in this study from the CALM sample. Note, task descriptions (except following instructions) are taken directly or paraphrased from [80] ([80]).

Cognitive Domain	Task Descriptions	Mean (SD) [Range]	Missing Data	Reference
Crystallized Ability (gc)	Numerical Operations (NO): Participants answered written mathematical problems that increased in difficulty.	16.1 (8.4) [0, 64]	9.94%	[92] ([92])
	Single-Word Reading (Read): Participants read aloud first a list of letters and then words that gradually increased in complexity. Correct responses required correctness and fluency.	83.2 (24.8) [1, 130]	2.48%	
	Spelling (Spell): Participants spelled words with increasing difficulty one at a time that were spoken by an examiner.	22 (9.2) [0, 49]	3.35%	
	Peabody Picture Vocabulary Test (Pea): Participants were asked to choose the picture (out of four multiple-choice options) showing the meaning of a word spoken by an examiner.	136.8 (31.6) [8, 215]	1.12%	[25] ([25])
Fluid Ability (gf)	Matrix Reasoning (MR): Participants saw sequences of partial matrices and selected the response option that best completed each matrix.	11.2 (5.6) [0, 28]	0.12%	[94] ([94])
Working Memory (WM)	Digit Recall (DR): Participants recalled sequences of single-digit numbers given in audio format.	24.6 (5.4) [7, 47]	0.5%	[3] ([3])
	Backward Digit Recall (BDR): Same as regular digit recall but in reversed order.	9.7 (4.4) [0, 25]	3.11%	
	Dot Matrix (Dot): Participants were shown the location of a red dot in a sequence of 4 × 4 matrices and had to recollect the location and order of these sequences.	18 (5.7) [2, 43]	0.75%	
	Mr. X (MrX): Participants remembered spatial sequences of locations of a ball held by a cartoon man rotated in one of seven positions.	9.3 (5.1) [0, 32]	1.24%	
	Following Instructions (FI): Participants carried out various sequences of actions (touch and/or pick up) based on props (a box, an eraser, a folder, a pencil, or a ruler) presented in front of them. By having participants undertake actions sequentially (do X “then” do Y), increasingly longer sequences were made which increased the difficulty. Scores denote total number of correct responses.	11.2 (4) [1, 33]	6.83%	[41] ([41])

## Data Availability

Due to privacy issues, we cannot share the data.

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
