# Peer review of "Bridging Brain and Cognition: A Multilayer Network Analysis of Brain Structural Covariance and General Intelligence in a Developmental Sample of Struggling Learners"

_jintelligence, 2021, doi:10.3390/jintelligence9020032_

Round 1

Reviewer 1 Report

The authors integrate cognitive abilities and grey- and white matter measures in a multilayer network model in an attempt to understand general intelligence in a developmental cohort of struggling learners. While the overall goal and methodology of the study are highly interesting, several key points warrant improvement – mainly pertaining to the interpretation of the results. A point-by-point response is provided below.

Introduction:

  • Relevance and aim are clear, but the introduction would benefit from more concise writing. Overall sentence structure/grammar needs additional attention, for example, please omit results from the introduction section (last paragraph).

Methods:

  • Differences in group sizes do exist for the behavioral and different brain outcomes, how do these differences in sample size impacts the outcome?
  • The cohort consists of “children [with] perceived difficulties…” – how and by whom are these difficulties assessed? And how do the participants of this study perform cognitively compared to their peers? Do they really score below what is to be expected.
  • How do the cognitive tests used in this study relate to more widely used tests for determining intelligence in children – e.g. the WISC.
  • Regarding the correlation plots (Figure 1): please move these plots to the results section. Also, I would like to see a correlation plot for CV and FA as these are included in the final multilayer network.
  • If not discussed in the text, the age plots (Figure 2) would be more appropriate in the Supplementary Material.

Discussion:

  • The interpretation of the findings stay superficial and need more elaboration. The integration of behavior and brain is the most interesting part of this study and it would be good to discuss specifically this integration. e.g. the “bridges” between the levels of organization – and what these finding mean for the research question asked as well as beyond the research question and how we look at the brain an cognition in general. What do the results tell us? As it stands, the multilayer network does not seem to add much or reveal anything additional to what the cognition network already tell by itself. Can the authors please comment on this?

Reviewer 2 Report

Thank you for the opportunity to review this study. Network analytic methods were used to model brain-behavior relations. Results suggest that cognitive and neural nodes play an intermediary role in relations between the brain and behavior. This study has many strengths. An impressive collection of variables was considered using complex network modeling. The description of the method and presentation of results are exemplary. My concerns relate to statements made in the Introduction section.

First, the authors suggest that the positive manifold emerges gradually from positive interactions among cognitive abilities. Mutualism is plausible and it is reasonable to frame the study around this assumption. However, evidence supporting mutualism is not so strong as to establish a “need" to conceptualize intelligence as a "complex dynamical system” (p. 2). Further research, including well-designed longitudinal studies, is needed to determine if such conceptualization is warranted. Thus, the statement regarding “need” should be tempered. For example, the authors might state that “in accordance with the mutualism model, general intelligence can be conceptualized as a complex dynamical system”.

Second, the authors claim that Schmank et al. (2019) reported that network models provided a better fit to intelligence data than do latent variable models. Schmank et al. (2021) note that results from direct comparisons of network and latent variable models were not reported in their 2019 article. Further, they caution readers to avoid making direct comparisons of these models using fit indices.

The authors do make some strong claims in the Discussion section given that results are based on cross-sectional data. However, this limitation is noted on page 21 and thus I do not have concerns with the Discussion section in its current form. I believe that this work represents a potentially impactful contribution to the literature and, pending revision, is deserving of publication.  

References

Schmank, C. J., Goring, S. A., Kovacs. K., & Conway, A. R. A. (2019). Psychometric network analysis of the Hungarian WAIS.” Journal of Intelligence 7(3): 21. https://doi.org/10.3390/jintelligence7030021

Schmank, C. J., Goring, S. A., Kovacs. K., & Conway, A. R. A. (2021). Investigating the structure of intelligence using latent variable and psychometric network modeling: A commentary and reanalysis. Journal of Intelligence 9: 8. https://doi.org/10.3390/jintelligence9010008

Reviewer 3 Report

In this paper the authors present a study that builds on previous findings of other researchers, to investigate in more detail individual defferences in cognitive ability using network theory and models (LASSO), by combining a network psychometrics approach with structural covariance networks derived from structural brain properties. I found the paper to be overall very well written and much of it to be well described. I felt confident that the authors performed careful and thorough field processing and have succesfully included an extended list of previous work and most of the state of the art literature that is also up to date. The authors have presented in a clear manner a descriptive analysis of previous works on this topic, stated the limitations and they are proposing a new methodology describing clearly the novelties and advantages of their method (section 2). I have found that they have succeeded in setting the purpose and goals of their research in a very clear manner and have made clear the differences of their work compared to other already proposed in the literature. The design of their methodology is very good, with solid scientific foundations, their results are well described and presented and scientifically justified (also after examination of the supplementary material) (Sections 3 and 4). 
In summary, I think this is a very rigorous paper that will make an important contribution to the literature and I propose it for as-is publication.

Author Response

Reviewer 3 did not have any suggestions for improvements to the manuscript. 

Round 2

Reviewer 1 Report

The paper has significantly improved. Most of the comments are adequately tackled. Just a minor thing, since we had some questions about the sample, would it be possible to add information about the cohort (and the differences between the groups) in the paper or in supplementary material? Or at least refer to the calm website in the paper. Besides that, no further questions!